# Transcriptome analysis of the eggs of the silkworm pale red egg ($re^p$-1) mutant at 36 hours after oviposition

**Meina Wu**[1,2]**, Pingyang Wang**[1,2,3]**, Mengjie Gao**[1,2]**, Dongxu Shen**[1,2]**, Qiaoling Zhao**[1,2]*

**1** School of Biotechnology, Jiangsu University of Science and Technology, Zhenjiang, Jiangsu, China, **2** The Sericulture Research Institute, Chinese Academy of Agricultural Sciences, Zhenjiang, Jiangsu, China, **3** Guangxi Zhuang Autonomous Region Research Academy of Sericultural Science, Guangxi, Nanning, China

* qlzhao302@126.com

**Data Availability Statement:** The transcriptome data has been deposited at NCBI under accession number PRJNA622872.

**Funding:** This work was supported by grants from the National Natural Science Foundation of China

## Abstract

The egg stage is one of the most critical periods in the life history of silkworms, during which physiological processes such as sex determination, tissue organ formation and differentiation, diapause and pigmentation occur. In addition, egg color gradually emerges around 36h after oviposition. The red egg mutant $re^p$-1, which was recently discovered in the $C_1$(H) wild-type, $C_1$(H) exhibits a brown egg color. In this study, the transcriptome of the eggs was analyzed 36h after oviposition. Between the $re^p$-1 mutant and the $C_1$(H) wild-type, 800 differentially expressed genes (DEGs) were identified, including 325 up-regulated genes and 475 down-regulated genes. These DEGs were mainly involved in biological processes (metabolic process, cellular process, biological regulation and regulation of biological process and localization), cellular components (membrane, membrane part, cell, cell part and organelle) and molecular functions (binding, catalytic activity, transporter activity, structural molecule activity and molecular transducer activity). The pathway enrichment of these DEGs was performed based on the KEGG database, and the results indicated that these DEGs were mainly involved in pathways in the following categories: metabolic pathways, longevity-regulating pathway-multiple species, protein processing in endoplasmic reticulum, peroxisome, carbon metabolism and purine metabolism. Further analysis showed that a large number of silkworm growth- and development-related genes and ommochrome synthesis- and metabolism-related genes were differentially expressed, most of which were up-regulated in the mutant. Our research findings provide new experimental evidence for research on ommochrome pigmentation and lay the foundation for further research on the mechanism of the $re^p$-1 mutant.

## 1. Introduction

The silkworm (*Bombyx mori*) is an economically important insect and a fully metamorphic lepidopteran model insect [1]. The entire life cycle involves 4 stages, including eggs, larvae,

(No. 31972616) (Qiaoling Zhao)and the Innovation Plan of Jiangsu Province (KYCX19_ 1656) (Meina Wu). The funders had no role in study design, data collection and analysis, decision to publish, or preparation of the manuscript.

**Competing interests:** The authors have declared that no ompeting interests exist.

pupae and moths. Diapause occurs in the silkworm egg stage. Therefore, the egg stage is one of the most important stages in the life cycle of the silkworm. Initially, the eggs appear light yellow, whereas diapause eggs gradually become gray-green, gray-purple or light brown. Other uncommon egg colors, such as red, white, brown, orange, and purple, are caused by mutations. Silkworm eggs are mainly composed of the egg shell and complex contents, including the yolk membrane, serosa, egg yolk, and embryo in addition to other components. Egg color arises from a number of expression events in the egg shell, serosa pigment, embryo, and yolk. However, ommochrome in the serosa is the determining factor [2]. Ommochrome is a tryptophan metabolite in insects that prevents tryptophan poisoning [3]. Ommochrome forms different colors by participating in redox reactions and is one of the main pigment substances in insects [4]. Tryptophan is converted into formylkynurenine via the opening of the pyrrole ring by a tryptophan pyrrolase (TRPO), after which kynurenine formamidase (KFase) causes a change in kynurenic, which is further transformed into 3-hydroxy-canine with the aid of kynurenine 3-hydroxylase ($K_3H$) ureine. Eventually, two molecules of 3-hydroxy-kynurenine are synthesized by phenoxazinone synthetase (PHS) to produce xanthommatin [5–8]. Ommochrome is mainly found in compound eyes and eggs. Additionally, it exists in the epidermis and central nervous system of silkworm larvae [9]. 3-Hydroxykynurenine in the blood in the hemolymph of the pupal stage enters the ovaries and compound eyes to affect egg color and adult compound eye color, respectively [10].

Egg color mutants are good model systems for investigating pigment metabolism. A large number of egg color mutants are maintained in the silkworm mutant library, and many new mutants are continually being discovered. The initial silkworm white egg (w-1) phenotype identified cannot produce ommochrome because the lack of kynurenine-3-hydroxylase leads to an accumulation of kynurenine, which cannot be converted into 3-hydroxy kynurenine [11]. In contrast, another mutant silkworm white egg (w-2) phenotype is yellow-white in the early stage and then develops a reddish color. Researchers have shown that the mutant form results from a mutation in a ATP-binding transporter. Dimers formed by ATP-binding cassette transporters and products encoded by the *white* gene are responsible for ommochrome transport [12]. Multiple alleles contribute to the third type of mutant silkworm egg phenotype (w-3), including the white egg worm ($w$-$3^{oe}$) mutant, which is caused by a single-base deletion in exon 2 of the *white3* gene [13]. The serosa cells of the red egg mutant (*re*) are red and contribute to the eggs turning dark red. Map-based cloning has proven that a new member of the transmembrane protein family causes the mutation [14].

The pale red egg mutant (pale red egg, $re^p$-1) is a new egg color mutant found in the silkworm breed $C_1(H)$. Its color gradually changes to pale red over approximately 36h to a color similar to that of the red egg mutant (*re*) albeit a different intensity (Fig 1). By hybridizing the $re^p$-1 mutant with the wild type p50 silkworm variety, the genetic model indicated that the pale red egg mutation was controlled by an autosomal recessive gene [15]. When the $re^p$-1 (female) mutant was crossed with the *re* mutation, or the *re* (female) mutant was crossed with the $re^p$-1 mutation, all of the F1 eggs were the same color as the female parent, whereas the F2 eggs were all red. The F3 egg color, however, segregated, with different degrees of red that were difficult to distinguish from red eggs and pale red eggs. Gene mapping showed that the key gene controlling the $re^p$-1 mutant is very close to the *MFS* (major facilitator superfamily) gene that controlled the *re* mutation (data not shown).

Because the egg color change starts 36h after oviposition, to understand the gene expression profile of the mutant at this time, eggs of the $re^p$-1 silkworm mutant laid within 36h were used as the experimental materials, and $C_1(H)$ as a control group. Fifty eggs from each silkworm strain were pooled for transcriptome analysis. Three experiments were repeated for each group, and the transcriptome data to NCBI with the login number of PRJNA622872.

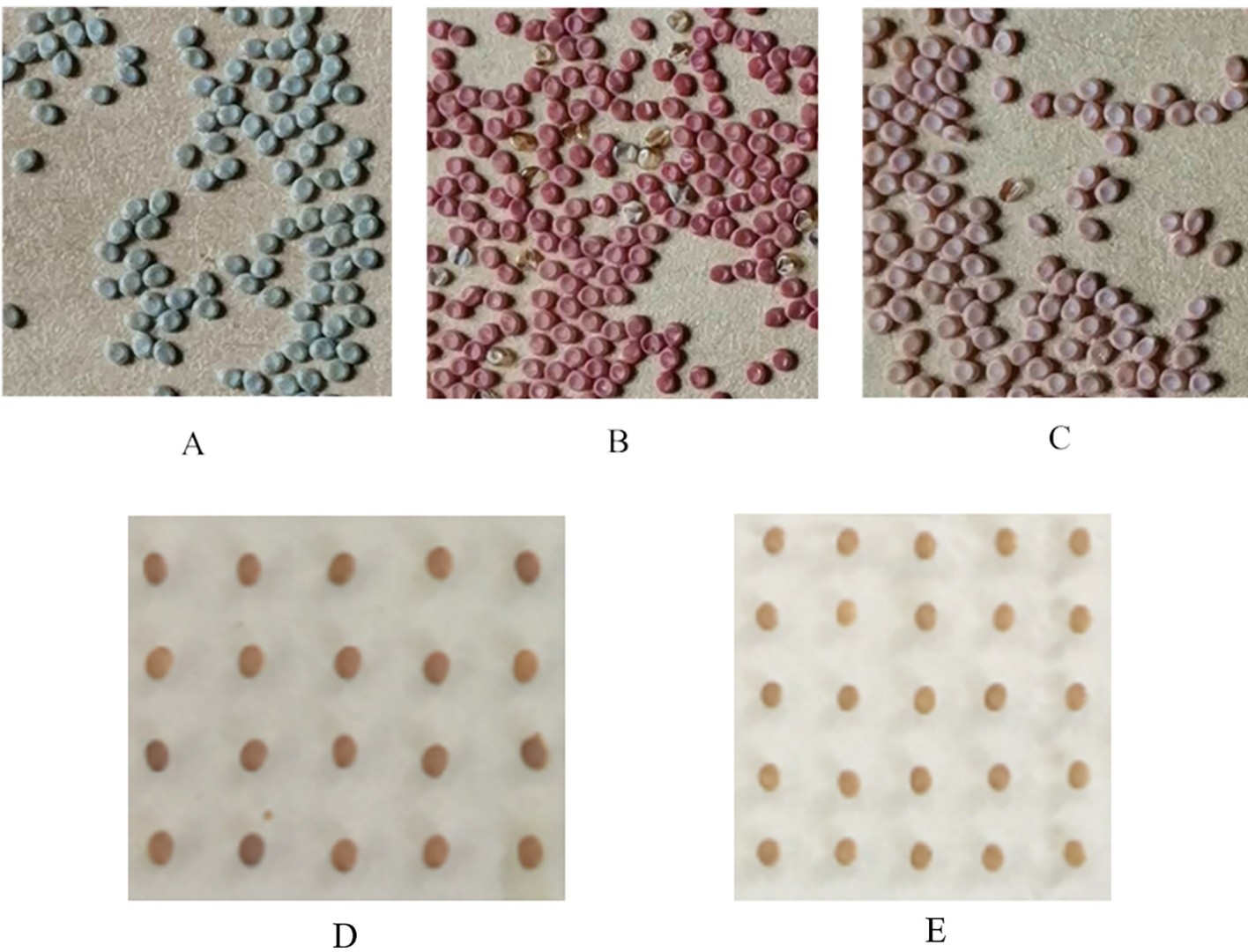

**Fig 1. Comparison of silkworm egg color.** (A) Normal type $C_1$(H). (B) Pale red egg ($re^p$-1) mutant. (C) Red egg ($re$) mutant. (D) Normal type $C_1$(H) after 36h of oviposition. (E) Pale red egg ($re^p$-1) mutant after 36h of oviposition. Finally, the eggs of normal type $C_1$(H) was black colors, the eggs of $re^p$-1 mutant were pale red colors, the eggs of $re$ mutant was red colors.

Differentially expressed genes (DEGs) were identified according to the sequencing results and further analyzed. At the same time, the expression levels of genes related to ommochrome synthesis and metabolism, egg development and structure and major genes associated with egg color mutants were analyzed for possible roles of these genes in the $re^p$-1 pale red egg mutant. The study provides a foundation for understanding the formation mechanism of $re^p$-1 mutants and as well as a theoretical basis for the study of the molecular mechanism controlling egg color formation.

## 2. Materials and methods

### 2.1. The silkworm strains

The $C_1$(H) and $re^p$-1 silkworm strains were provided by the Chinese Academy of Agricultural Sciences (Zhenjiang. China). Larvae were reared on fresh mulberry leaves in an environment

with a 12-h light/12-h dark cycle at 25 ± 1˚C and 70–85% relative humidity. After 36h of oviposition, 50 eggs of each silkworm strain were pooled for RNA-Seq and quantitative analysis. Each assay was repeated three times. Eggs at 12h, 24h and 36h after oviposition were used to examine the expression of genes of interest. This experiment was approved by the Jiangsu University of Science and Technology Animal Care and Use Committee.

## 2.2. RNA extraction, library construction and RNA-Seq

RNA was extracted using RNAiso Plus (TaKaRa, China) and dissolved in RNase-free water. RNA purity was assessed by using a NanoDrop 1000 microspectro- photometer (Thermo, USA). Then total mRNA treated with RNase was enriched using oligo (dT) magnetic beads. The next step was to shear mRNA into short fragments by adding hybridization interruption reagents. Double-stranded cDNA was synthesized using random six-base primers and the interrupted mRNA as the template. The purified double-stranded cDNA was end repaired, and a tail was added and connected with a sequencing connector. PCR amplification was carried out to generate cDNA library. The library constructed in the previous step was loaded onto the sequencing chip of an Illumina HiSeq TM 2000 sequencer for sequencing (NanJing Decode Genomics). Finally, 150-bp paired end reads data of the processing group and the control group library were obtained.

## 2.3. Sequence assembly

After filtering out low quality reads and sequence reads containing connectors or poly-A / T tails, clean reads were assembled by using the short reads assembler Trinity-v2.8.4. All experiments were performed with Trinity using parameters: minimum contig length of 100 bases and average fragment length of 300 bases [16]. The unigenes from two sample groups were gathered together and clustered using the TGI clustering tool [17]. For functional annotation, BLASTX (E-value cut-off = $10^{-5}$) was used to retrieve single gene sequences from various protein databases including Nr, Swiss-Prot, COG and KEGG [18], and BLASTN was used to search the nucleotide database (E-value $< 10^{-5}$). The coding sequence (CDS) was extracted from unigenes by searching BLAST results and translated into peptide sequence. When unigene failed to generate a corresponding BLAST hit, the sequence direction was determined through ESTScan software 2.1 [19]. In addition, a Gene Ontology (GO) Association was performed using the Nr database according to BLASTX, and each gene was annotated with Blast2GO [20, 21].

## 2.4. Quality control of RNA-Seq data

Quality control analysis was performed on the raw data, which were filtered to obtain clean reads. The filtering rules were as follows: removal of reads with adapter sequences at first; then removal of reads with N bases (N means that the base information cannot be determined); finally removal of reads with low quality (in which the number of reads with a quality value of Qphred ≤ 5 is more than 50% of the total reads). At the same time, the distribution of base quality values, the base composition and the average GC content of the original sequence data were analyzed through ESTScan software 2.1 to ensure that the sequencing data met the requirements for subsequent analysis.

The clean reads were compared with the reference genome (NCBI, https://www.ncbi.nlm.nih.gov/)), the statistical reads in the distribution of different segments of the genome and the results of the second quality analysis, including read coverage uniformity, insert sequence fragment length and sequencing saturation analyses, to ensure that the reads met the requirements for subsequent gene expression data and DEG analyses.

## 2.5. Analysis of differentially expressed genes

The number of reads obtained from the genome was counted, and the RPKM (reads per kilobase per million reads) method using RSEM program of Trinity software-v2.8.4 was used to standardize the expression level of each gene [22]. The boxplot and density distribution of the RPKM values and the correlation analysis of each sample were performed to compare the differences in gene expression levels among different samples according to the overall level of gene expression. Then, an FDR≤0.05 (FDR: false discovery rate) was used as the threshold for the screening of DEGs [23]. GO and KEGG pathway enrichment analyses were performed for the differentially expressed genes encoding proteins. The expression levels of genes related to ommochrome synthesis and metabolism, genes related to egg development and structure and major genes of the egg color mutants were analyzed simultaneously to elucidate the possible roles of these genes in *re^p*-1 pale red egg mutants.

## 2.6. Quantitative Reverse Transcription PCR (qRT-PCR)

Eggs were produced by $C_1$(H) and *re^p*-1 mutant silkworms for 36h. For each sample, 50 eggs were pooled. The RNAiso plus method was used to extract the total RNA, followed by treatment with DNase, and cDNA was obtained using an M-MLV Reverse Transcriptase (RNase H-) kit. The cDNA was diluted to 100 ng/μl as a template for qRT-PCR. All qRT-PCR reactions were performed in a 20 μl volume containing 1μl specific primers (10 μM), 1μl cDNA, 10 μl NovoStart®SYBR qPCR SuperMix (Novoprotein, Nanjing, China) and $H_2O$ in a Light-Cycler® 96 quantitative PCR instrument. The reaction conditions were as follows: predenaturation at 95˚C for 10 min, 40 three-step cycles including denaturation at 95˚C for 10 s, annealing at 58˚C for 10 s, and extension at 72˚C for 10 s (58˚C to 95˚C in an increment of 0.5˚C every 5 s). Finally, the melting curve was generated. The relative expression of each gene was calculated in relation to the average values for two housekeeping genes, RPL-3 (*BGIBMGA013567*) and GAPDH (*BGIBMGA007490*). The $2^{-\Delta\Delta Ct}$ method [24] was used to examine differential expression of the genes of interest. qRT-PCR expression in the wild-type was compared with *re^p*-1 to verify the accuracy of the DEG data.

## 3. Results

### 3.1. General information for RNA-Seq

Eggs that hatched after 36h were used for RNA-seq. raw reads were obtained from the wild-type $C_1$(H) group (55.72 ± 6.22), and Mega clean reads (55.54 ± 6.19) were obtained following quality control. Similarly, raw reads (64.08 ± 7.50) and Mega clean reads (63.88 ± 7.50) were obtained from the *re^p*-1 mutant. The ratios of clean reads and raw reads were 99.67% and 99.68%, respectively. The GC content of the reads for both the wild-type $C_1$(H) group and the *re^p*-1 mutant was approximately 43%, and that of adapter reads was approximately 3% (Table 1).

The analysis of distribution of the base mass values of the sequencing data (S1A Fig, S1D Fig), the distribution of the composition of each base (S1B Fig, S1E Fig) and the distribution of the average GC content (S1C Fig, S1F Fig) showed that the base mass was good; the base composition was uniform; and the GC content was stable, thus satisfying the requirements for subsequent analysis. In addition, the Q20 values of the raw reads and clean reads of $C_1$(H) and *re^p*-1 were greater than 95%. The Q30 values of the raw reads and clean reads of $C_1$(H) and *re^p*-1 were greater than 90% (Table 1), thus meeting the requirements for subsequent analysis.

**Table 1. General information of RNA-seq data.**

| Category | Parameter | Wildtype | $re^p$-1 mutant |
|---|---|---|---|
| Raw data | Total Reads (Megabases) | 55.72 ± 6.22 | 64.08 ± 7.50 |
| | Total bases (Gigabases) | 8.35 ± 0.92 | 9.61 ± 1.13 |
| | Q20 (%) | 97.56 ± 0.05 | 97.41 ± 0.06 |
| | Q30 (%) | 93.38 ± 0.11 | 93.08 ± 0.13 |
| | Adapter reads (%) | 3.13 ± 0.09 | 3.03 ± 0.09 |
| | GC (%) | 43.33 ± 0.09 | 43.32 ± 0.20 |
| Clean data | Total Reads (Mega) | 55.54 ± 6.19 | 63.88 ± 7.50 |
| | Total bases (Gigabit) | 8.31 ± 0.92 | 9.56 ± 1.12 |
| | Q20 (%) | 97.76 ± 0.05 | 97.62 ± 0.05 |
| | Q30 (%) | 93.61 ± 0.10 | 93.31 ± 0.12 |
| | GC (%) | 43.27 ± 0.09 | 43.27 ± 0.20 |
| | Clean data/Raw data (%) | 99.67 | 99.68 |
| Quality Control | Clean Read Q20(%) ≥ 95 (Y or N) | Y | Y |
| | Clean Read Q30(%) ≥ 90 (Y or N) | Y | Y |
| | Clean Reads ≥ 10M (Y or N) | Y | Y |
| | Gene Unique Mapping Ratio(%) ≥ 80 (Y or N) | Y | Y |
| Genome-Mapping | Total Mapped Reads (%) | 87.09 ± 0.01 | 87.46 ± 0.01 |
| | Unique Match (%) | 83.44 ± 0.01 | 83.90 ± 0.01 |
| | Multi-position Match (%) | 2.72 ± 0.01 | 2.69 ± 0.01 |
| | Total Unmapped Reads (%) | 12.91 ± 0.01 | 12.54 ± 0.01 |
| Gene expression | Total Genes | 15884 | 15884 |
| | Expressed genes (%) | 12403 (78.08%) | 12488 (78.62%) |
| | Both expressed genes (%) | 12007 (75.59%) | |
| | Not expressed genes (%) | 3481 (21.91%) | 3396 (21.38%) |
| | Neither expressed genes (%) | 3000 (18.89%) | |
| | Expressed only in $C_1$(H) (%) | 396 (2.49%) | — |
| | Expressed only in $re^p$-1 (%) | — | 481 (3.03%) |

## 3.2. Read mapping analysis

The clean reads obtained from RNA-seq were mapped with the NCBI reference genome seventy-six. and the accession number is ASM15162v1. In $C_1$(H) and $re^p$-1, 87.09% and 87.46% of the reads were mapped to the reference genome. A total of 83.44% and 83.90% of the reads, respectively, were mapped to the single-copy reference sequence, and 2.72% and 2.69% of the reads were mapped to the multicopy reference sequence. The number of clean reads exceeded 55 megabases, reaching the minimum requirement of 10 megabases, and a unique gene mapping ratio reached 80%.

The distribution statistics of the reads in different regions of the genome showed that more than 87% of the reads matched to a CDS, and some of the reads matched intron and gene intervals, possibly due to the presence of unknown genes and alternative splicing (S2A Fig, S2E Fig) The coverage uniformity analysis of the reads showed that the coverage of 10 windows at the 5' end was limited. The coverage of 10 to 60 windows was best. The coverage of the 3' end gradually decreased (S2B Fig, S2F Fig). The analysis of insert length showed that the average distances between two reads in $C_1$(H) and $re^p$-1 were 11 BP and 14 BP, respectively, which showed that the selected length could basically cover two reads and achieved the maximum utilization of the sequence (S2C Fig, S2G Fig). The saturation analysis indicated that saturation was reached when the sequencing percentage was approximately 40%, and all of the

sequencing requirements were met (S2D Fig) Therefore, mapping results met the requirements for subsequent analysis.

### 3.3. General gene expression information

In total, we analyzed the expression levels of 15884 gene (S2 Table). Among these genes, 12403 were expressed in wild-type $C_1$(H), accounting for 78.08% of the total, while 12488 (78.62%) were expressed in the *re^p*-1 mutant (3 replicates could be detected). A total of 12007 (75.60%) genes were expressed in both $C_1$(H) and *re^p*-1 (6 replicates could be detected); 396 (2.50%) genes were expressed in $C_1$(H) but not in *re^p*-1; and 481 genes were expressed in *re^p*-1 but not in $C_1$(H), accounting for 3.03% of the total. Finally, 3000 genes were not expressed in $C_1$(H) and *re^p*-1 (could not be detected in at least one of the six samples), accounting for 18.89% of the total (Table 1).

The correlation analysis results for sample expression showed that the correlations between the six samples were very close; the correlation coefficient was close to 1 (S3A Fig); the box plot analysis results showed that the repeatability of the gene expression results of the six samples was good. The maximum value, the upper quartile, the median value, the lower quartile and the minimum value showed good uniformity among the six samples (S3B Fig). A horizontal density distribution diagram of gene expression was generated. Genes showing an expression value of approximately 10 were the most abundant (the first peak). Genes with an expression value of approximately 0.1 were also highly expressed (the second peak) (S3C Fig).

### 3.4. Analysis of differentially expressed genes

An FDR $\leq$ 0.05 was used as a threshold for screening differentially expressed genes indicated in S3 Table. We identified 800 differentially expressed genes (S3 Table). Compared with the $C_1$(H) wild-type, 475 genes in the *re^p*-1 mutant were down-regulated, and 325 genes were up-regulated (S3D Fig–S3F Fig). GO cluster analysis was carried out on the differentially expressed genes. Among these genes, the identified biological processes mainly included the following terms: metabolic process (103 up-regulated genes and 115 down-regulated genes), cellular process (85 up and 115 down), biological regulation (27 up and 66 down), regulation of biological process (24 up and 59 down) and localization (22 up and 46 down). The identified cellular component terms mainly included the following: membrane (67 up and 110 down), membrane part (60 up and 96 down), cell (62 up and 80 down), cell part (61 up and 80 down) and organelle (41 up and 35 down). The molecular function terms were mainly related to binding (100 up and 142 down), catalytic activity (104 up and 133 down), transporter activity (8 up and 28 down), structural molecule activity (6 up and 7 down) and molecular transducer activity (3 up and 12 down) (Fig 2, S4 Table). Based on a difference multiple greater than 2, that was |log2(FOLD change) | > 1, 114 genes were up-regulated and 120 were down-regulated, and the 234 differentially expressed genes were mainly related to binding proteins (56) and catalytic activity (64).

These differentially expressed genes were analyzed by using the KEGG database to evaluate pathways, and 218 genes were clustered into 105 pathways. The top six clustered genes belonged to the following categories: metabolic pathways (89), longevity regulating pathway—multiple species (16), protein processing in the endoplasmic reticulum (16), peroxisome (14), carbon metabolism (14) and purine metabolism (14) (Fig 3, S5 Table). Among the metabolic pathways, a large number of amino acid metabolic pathways showed changes, including those of serine, cysteine, methionine, leucine, lysine, arginine, proline, tyrosine, polyamines, p-aminobutyric acid and glutathione.

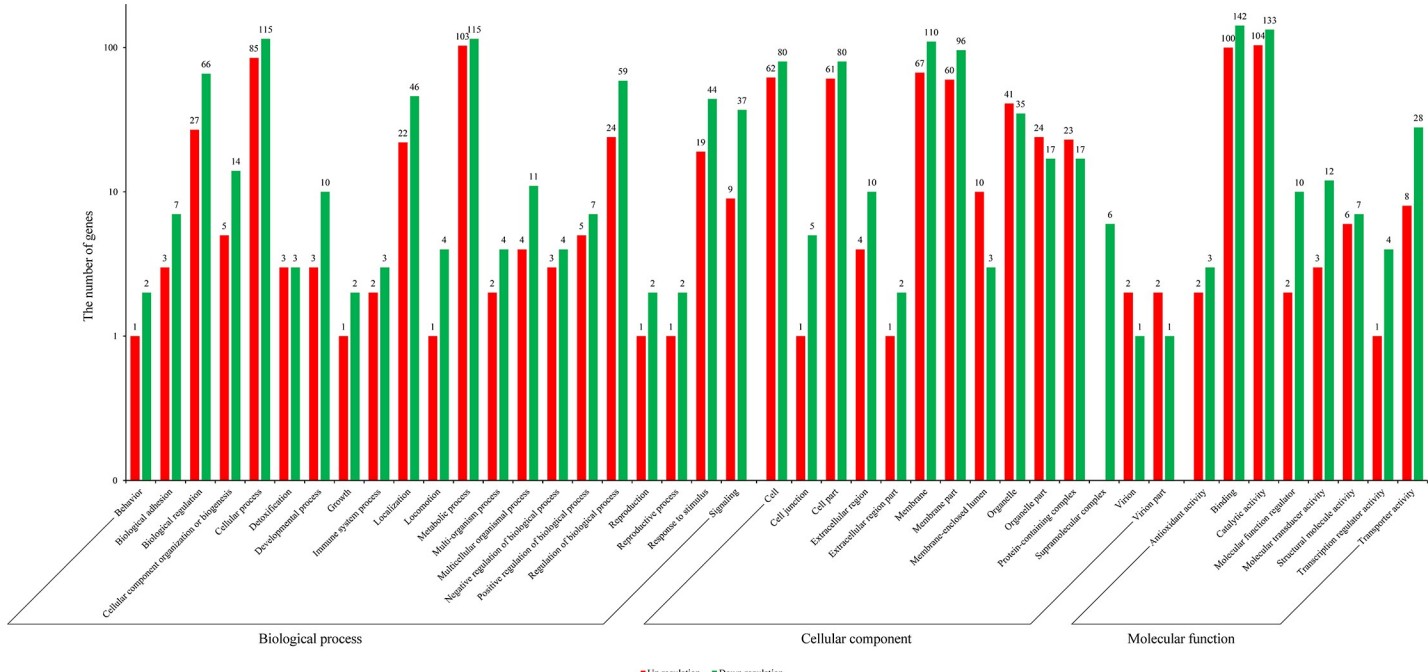

**Fig 2. Gene ontology enrichment of differentially expressed genes.** These DEGs were mainly involved in biological process, cellular component and molecular function.

## 3.5. Verifying the accuracy of the RNA-Seq data by qRT-PCR

To verify the accuracy of the RNA-seq data, we analyzed the expression of three housekeeping genes, and selected certain genes for qRT-PCR analysis, including representative differentially expressed genes, pigment synthesis genes, metabolism-related genes, egg shell protein genes, yolk membrane protein genes, and major genes affected by egg color mutants (Table 2, S1 Table).

Three housekeeping genes, *bmactin3* (fold-change = 1.07). Glyceraldehyde-3-phosphate dehydrogenase (fold-change = 0.81) and ribosomal protein L3 (fold-change = 0.97), showed fold-changes in all databases that were approximately equal to 1, providing initial verification that the RNA-seq data were accurate. Twelve up-regulated genes and twelve down-regulated genes with fold-change differences greater than 2 were randomly selected for qRT-PCR validation. The results for 20 out of 24 DEGs were identical to the qRT-PCR results, whereas the qRT-PCR results for 2 out of 4 genes were inconsistent with the DEG results. However, the fold-change difference was not significant ($0.5 <$ fold-change $< 2$). Therefore, the results for 22 of the DEGs genes were consistent with the qRT-PCR results. The accuracy was 91.67%, which indicated that the DEG results were reliable. The expression levels of select genes are shown in Fig 4.

## 4. Discussion

### 4.1. Differentially expressed genes based on RNA-Seq data

Serotonin is a widespread distribution in the insect nervous system [25]. It affects sensory processing, information coding and behavior [26]. The glutamate-gat chloride channel is only found in invertebrates, mediating fast inhibitory neurotransmission [27]. The genes encoding these two proteins involved in the nervous system were down-regulated (fold-change = 0.49)

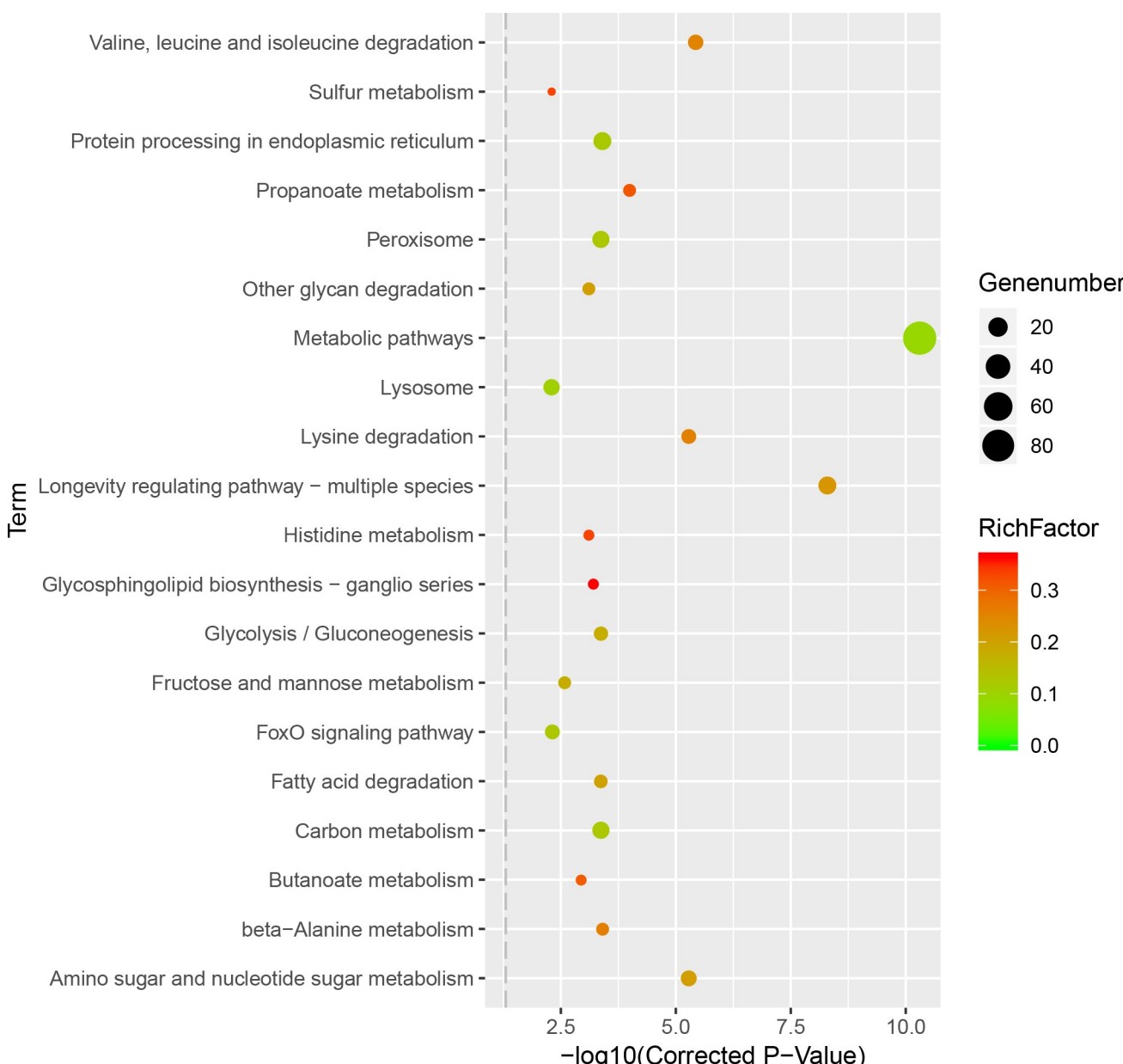

**Fig 3. Scatter plot of KEGG pathway enrichment.** The enrichment factor is the ratio of differentially expressed gene numbers annotated in this pathway term to all gene numbers annotated in this pathway term. The larger the enrichment factor meant the greater intensiveness.

**Table 2. Genes affected by known egg color mutants.**

| Mutant | Mutant gene | Gene products | References |
|---|---|---|---|
| *pe* | *Bmcardinal* | Phenoxazinone synthase | [8] |
| *re* | *Bmre* | MFS transporter | [14] |
| *w-1* | *BmK3H* | Kynurenine 3-hydroxylase | [11] |
| *w-2* | *Bmw-2* | Scarlet (ABC transporter) | [12] |
| *w-3*<sup>oe</sup> | *Bmwh3* | White (ABC transporter) | [13] |

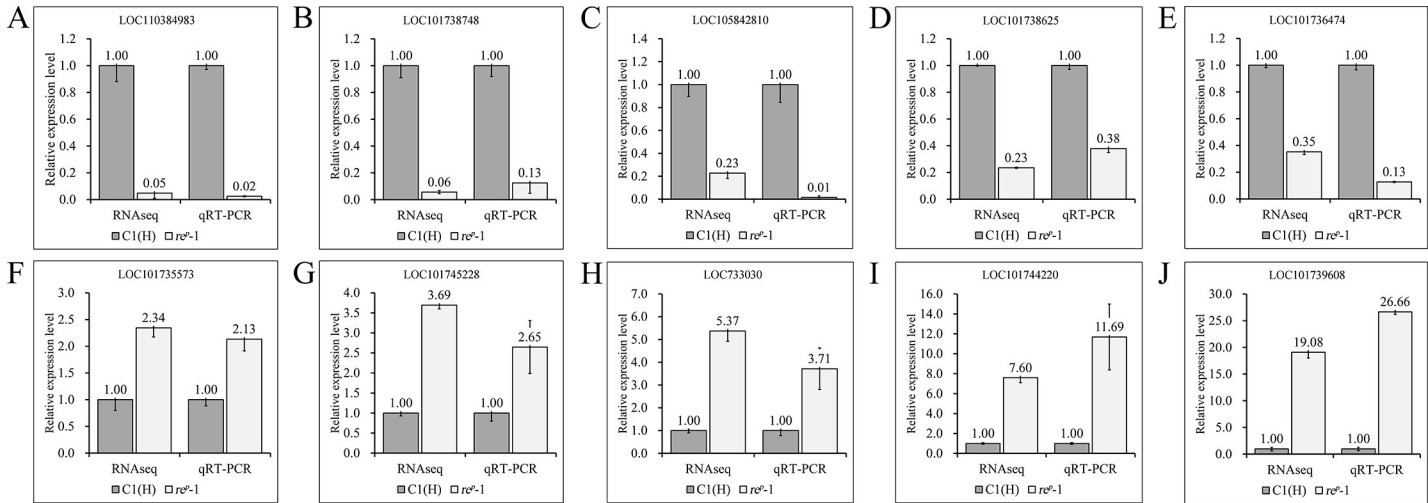

**Fig 4. Relative expression of partial genes verified by qRT-PCR.** The LOC number represented the gene ID in NCBI. All genes were closely consistent between RNA-seq and qRT-PCR, which indicated that the RNA-Seq data of RNA-seq are reliable.

and significantly up-regulated (fold-change = 11.69), respectively, suggesting that the development of the nervous system might be affected in the mutant. Troponin plays a role in muscle contraction [28]. Dynein plays a role in the movement of flagella and cilia [29]. The genes encoding these two proteins are related to motility and were up-regulated in the mutant (fold-change = 3.71 for troponin, while dynein was expressed only in the mutant). These results indicate that material transport and metabolism in the mutant are relatively similar. Other up-regulated genes among the genes involved in the metabolism of these substances included those encoding a key enzyme in the urea cycle, arginine succinate lyase, was also up-regulated (argininosuccinate lyase, fold-change = 16.83) [30], the Krebs cycle (tricarboxylic acid cycle) component isocitrate dehydrogenase (isocitrate dehydrogenase, fold-change = 3.28) [31], a protein with a role in cell adhesion (cytoadherence) (integrin, fold-change [32] = 7.48), the signal transduction-related enzyme serine-threonine protein kinase (serine/threonine—protein kinase, fold-change = 4.89) [33, 34] and Phosphoglyceryl acyltransferase (old-change = 26.66) involved in glycerolipid synthesis [35], further confirming the similar cellular activity and material metabolism of the mutant.

## 4.2. Differentially expressed genes involved in amino acid metabolism

KEGG pathway clustering analysis of the differentially expressed genes revealed 89 DEGs clustering to metabolic pathways. The analysis revealed many amino acid metabolic pathways with gene expression differences (S5 Table), including the serine, betaine, cysteine, arginine, proline, polyamine and glutathione biosynthesis pathways, methionine, leucine, lysine and tyrosine degradation pathways and aminobutyric acid metabolic pathways. Amino acids are not only the basic substances necessary for biological activities but are also the components of basic molecules such as proteins, which are involved in almost all physiological processes. Abnormal metabolism will affect the growth and development of the silkworm.

## 4.3. Differentially expressed genes related to ommochrome synthesis

Ommochrome is the key substance responsible for the formation of egg color. 3-Hydroxykynurenine, which is synthesized from tryptophan through a series of catalytic reactions, is

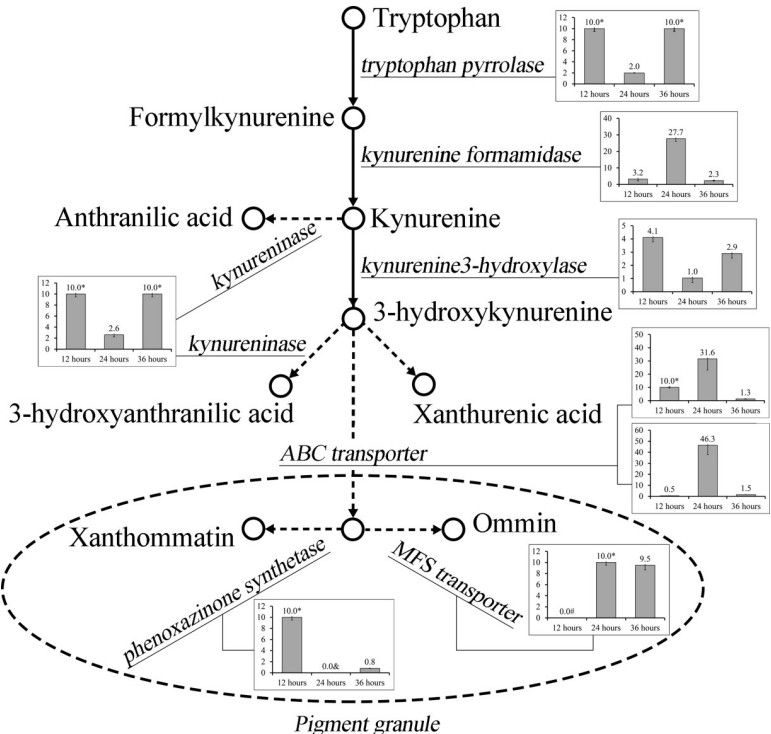

**Fig 5. Ommochrome synthesis pathway.** Ommochrome is synthesized through a series of enzymatic reactions employing tryptophan as the substrate. Most genes encoding these enzymes were up-regulated in the *re^p*-1 mutant especially at 24h after oviposition. The vertical axis is defined as the ratio of expression of *re^p*-1 mutant / expression of wildtype. If the ratio> 2 or <0.5, it indicates that there are differences in expression quantity.

transported to pigment particles by the ABC transporter heterodimer. Then, it is transformed via oxidative condensation to form purple ommin pigment and red-brown xanthommatin pigment [10] (Fig 5). In silkworms the scarlet protein encoded by the *Bmw-2* gene and the white protein encoded by the *Bmwh3* gene form a white/scarlet dimer, which transports 3-hydroxy-kynurenine, an ommochrome precursor, to pigment granules [12, 13]. The *Bmcardinal* gene encodes phenoxazinone synthase, which catalyzes the synthesis of xanthommatin. The other 3-hydroxykynurenine is used to synthesize ommin, which plays a key role in red egg mutants. The red egg mutant gene *MFS* (major facilitator superfamily) is related to the formation of ommin [14]. Kynurenine 3-hydroxykynurenine can simultaneously be hydrolyzed to produce anthranilic acid and 3-hydroxy-2-aminobenzoic acid via the action of the kynureninase gene *bmkynu* [36] (Fig 5).

The quantitative analysis of gene expression in the ommochrome synthesis and metabolism pathways at 12h, 24h and 36h after oviposition showed that the expression levels of most genes were up-regulated in the *re^p*-1 mutant. Compared to the $C_1(H)$ wild-type, there was a trend of increasing expression levels in eggs in the 12h postpartum period. Expression was significantly increased at 24h and tended to return to a level similar to that in the wild-type at 36h (Fig 5). According to the analysis of gene expression, the main effector gene of the *re^p*-1 mutant caused changes in gene expression in the ommochrome synthesis pathway. A series of genes were up-regulated in the mutant, causing more ommochrome to be produced. In relation to ovary coloration and the phenotypic changes in mutant pale red eggs, the changes in gene expression at different developmental stages were analyzed. The genes related to egg pigment synthesis began to be expressed at high levels at 12h after oviposition, reaching their highest expression

at 24h, after which their expression began to decrease. At 36h, ommochrome synthesis tended to be complete. Egg color began to appear, and gene expression was reduced to the lowest level. In particular, the expression levels of the two ABC transporters responsible for transporting 3-hydroxykynurenine to pigment granules (white and scarlet) reached a maximum at 24h. The results showed that 24h after oviposition may be the key period of ommochrome synthesis in silkworm eggs.

## 4.4. Expression analysis of key genes affected in several egg color mutants

Silkworm eggs represent not only the end of life but also the beginning of a new cycle. Egg diapause is useful for surviving in harsh environments, and the egg stage is therefore a very important developmental period. Egg color is a critical physiological characteristic of silkworm eggs. Normal silkworm eggs are gray-green, gray-purple or light brown. Many different egg color mutants have been identified through silkworm breeding. These egg color mutants are essential for studying silkworm egg pigment deposition. The color mutations and mutant genes identified thus far are shown in Table 2.

The mutant egg color genes that were identified were all related to the ommochrome synthesis pathway. Quantitative analysis showed that these genes were up-regulated in the pale red egg mutant. Most of the genes were up-regulated at 24h after oviposition (Fig 5). The results indicated that the egg color mutants largely result from mutations in the ommochrome synthesis pathway. The white egg mutant is unable to complete ommochrome synthesis because of gene mutations during the early stage of the synthesis pathway. However, compared to the white egg mutant, egg color mutations are caused by later gene mutations that result in different egg color phenotypes according to different ratios of the mixture of ommin (purple) and xanthommatin (red-brown).

## 4.5. Differentially expressed genes encoding structural proteins of eggs

Insect eggs are mainly composed of the shell and its contents, including the yolk membrane, protoplasm, vitellus, cell nucleus and other structures. The membrane protoplasm as noncellular structure is located close to the egg shell. The vitellus is very important because of its complex components, which provide nutrition for the development of the egg. The nucleus is responsible for the storage of genetic information. The egg shell is a complex protective structure on the surface of the egg with the physiological function of protecting embryos, gas exchange in the eggs [37]. The egg shell protein is the main component of the egg shell. A total of 96% of the dry weight of the silkworm is contributed by the egg shell protein. More than 100 egg shell proteins have been identified in silkworms to date [38, 39]. According to RNA-Seq data analysis, eight chorion-related genes were expressed. However, there was no significant difference in the expression of these genes ($|\log2(\text{fold-change})| < 0.5$). The results are consistent with egg shell construction being completed 36h after oviposition. Only a few egg shell proteins were expressed. The mutant genes of the pale red egg mutant did not affect the regulation and expression of egg shell proteins (S6 Table). The yolk protein precursor is mainly derived from the female-specific hemolymph protein vitellogenin (Vg) of the adipose body. After synthesis by the adipose body, vitellogenin is transferred to the ovary through the hemolymph, which is absorbed by oocytes and deposits vitellogenin in the egg to provide nutrients and functional substances for embryonic development [40–42]. By analyzing the expression of the vitellogenin gene, we found that the expression of all vitellogenin genes was up-regulated in the light red egg mutants except for one gene (101746088).

The skin an essential organ for the avoidance of harm, growth reproduction and adaptation to complex environments in insects [43, 44]. In the insect genome, cuticular protein genes

(CPG) are characterized by a number of structural features. There are more than 200 genes encoding CPGs in the silkworm genome [43, 45]. Approximately 30 CPGs were expressed in the 36h silkworm eggs. However, there was no significant difference between the wild-type and the mutants. In conclusion, the expression of genes encoding silkworm egg structural proteins was not significantly affected apart from some genes encoding vitellogenin.

## 5. Conclusion

In this study, the silkworm eggs of the light red pale egg mutant *re*$^p$-1 at 36h after oviposition were used as the experimental group, and those of the $C_1$(H) wild-type were used as the control group for RNA-seq data analysis. The qRT-PCR results verified the accuracy of the RNA-seq data. A total of 800 DEGs were identified by RNA-seq. Compared with $C_1$(H), 475 genes were down-regulated and 325 genes were up-regulated in *re*$^p$-1. It was found that the expression of genes related to the growth and development of silkworms and ommochrome synthesis and metabolism changed significantly. The overall trend was that the expression level was up-regulated in the mutants. At the same time, numerous DEGs related to amino acid metabolism pathway were found.

## Supporting information

**S1 Fig. Quality control of sequencing base.** (A) Base quality value distribution in wildtype $C_1$(H). (B) Base composition distribution in wildtype $C_1$(H). (C) Average GC content distribution in wildtype $C_1$(H). (D) Base quality value distribution in mutant *re*$^p$-1. (E) Base composition distribution in mutant *re*$^p$-1. (F) Average GC content distribution in mutant *re*$^p$-1.
(TIF)

**S2 Fig. Quality control of genome alignment.** (A) Distribution of reads in different regions of genome the wild type genome. (B) Homogeneous analysis of read coverage in wild type. (C) Length analysis of insert fragment in wildtype. (D) Saturation analysis in wild type. (E) Read distribution in different regions of the mutant. (F) Homogeneous analysis of read coverage in the mutant. (G) Length analysis of insert fragments in the mutant. (H) Saturation analysis in the mutant.
(TIF)

**S3 Fig. Analysis of general expression and differentially expressed genes.** (A) Expression correlation analysis between different samples. (B) Box plots of gene expression level. (C) Density graphic of gene expression level. (D) Cluster analysis of differentially expressed genes. (E) MA plot of differentially expressed genes. (F) Volcano Plot of differentially expressed genes.
(TIF)

**S1 Table. Primers.** Expression levels and descriptions of differentially expressed genes.
(XLSX)

**S2 Table. Read number and RPKM values of all genes.**
(XLSX)

**S3 Table. Differentially expressed genes.**
(XLSX)

**S4 Table. Gene ontology analysis of differentially expressed genes.**
(XLSX)

**S5 Table. KEGG pathway enrichment.**
(XLSX)

**S6 Table. Genes encoding chorion. Cpgs (Cuticular protein gene). Vitellogenin and ommo-chrome synthesis related enzymes.**
(XLSX)

## Author Contributions

**Data curation:** Meina Wu, Mengjie Gao.

**Funding acquisition:** Qiaoling Zhao.

**Investigation:** Meina Wu, Qiaoling Zhao.

**Methodology:** Qiaoling Zhao.

**Software:** Pingyang Wang.

**Validation:** Meina Wu, Pingyang Wang, Mengjie Gao, Dongxu Shen, Qiaoling Zhao.

**Writing – original draft:** Meina Wu, Pingyang Wang, Dongxu Shen.

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
