## [Decision Letter · Decision Letter 0]

25 Mar 2020

PONE-D-20-03586

Transcriptome analysis of the eggs of the silkworm pale red egg (rep-1) mutant at 36 hours after oviposition

PLOS ONE

Dear professor Zhao,

Thank you for submitting your manuscript to PLOS ONE. After careful consideration, we feel that it has merit but does not fully meet PLOS ONE’s publication criteria as it currently stands. Therefore, we invite you to submit a revised version of the manuscript that addresses the points raised during the review process. Reviewer 1, in particular, had questions regarding the nomenclature used regarding the term rep-1, which would suggest that the mutant examined is a derivative of the re mutant. If so, this was not fully demonstrated. Reviewer 1 also had a number of suggestions regarding the inclusion of data/images describing the mutant phenotype relative to WT and/or other mutants that would greatly strengthen the manuscript.

In addition, I had a number of comments/suggestions for you to consider.

1) As currently written, the Methods section lacks sufficient details (and references) to allow for replication. I recommend looking at the relevant sections of the following papers for examples on the details needed for sequencing/quality control/quantification of the Illumina data.

Thoma et al Front Ecol Evol 2019 “Transcriptome Surveys in Silverfish Suggest a Multistep Origin of the Insect Odorant Receptor Gene Family” https://doi.org/10.3389/fevo.2019.00281

Dou et al PLOS ONE 2019 “Pheromone gland transcriptome of the pink bollworm moth, Pectinophora gossypiella: Comparison between a laboratory and field population” https://doi.org/10.1371/journal.pone.0220187

Yang et al PLOS ONE 2018 “Transcriptome analysis in different developmental stages of Batocera horsfieldi (Coleoptera: Cerambycidae) and comparison of candidate olfactory genes” https://doi.org/10.1371/journal.pone.0192730

Kozma et al PLOS ONE 2020 “Comparison of transcriptomes from two chemosensory organs in four decapod crustaceans reveals hundreds of candidate chemoreceptor proteins” https://doi.org/10.1371/journal.pone.0230266

2) Similarly, qPCR methods and data should include amplification efficiencies (see Bustin et al Clin Chem 2009 The MIQE Guidelines: Minimum Information for Publication of Quantitative Real-Time PCR Experiments).

3) Related to Reviewer 1 comments re the nomenclature used. If rep-1 is a derivative of the re mutant, why was the re mutant not included in the analyses?

4) It is not clear if qualitative differences were examined. Relatedly, since this is referred to as rep-1, is the same/similar mutation in the MFS transporter reported in the original Osanai-Futahashi manuscript present? Line 357 states that the mutant is “unable to complete ommochrome synthesis because of gene mutations during the early stage..”. What are these mutations? As written, it suggests that the mutations are qualitative. If not, this should be re-phrased so that is more clear that the phenotype is driven by quantitative differences in transcript expression, which could indicate potential changes to transcription factors. If so, were these examined?

5) Indicate in Figure 7 the genes that differ from WT, perhaps by addition of colors or some other type of marker. It is not clear in the current version where the differences occur.

We would appreciate receiving your revised manuscript by May 09 2020 11:59PM. To enhance the reproducibility of your results, we recommend that if applicable you deposit your laboratory protocols in protocols.io, where a protocol can be assigned its own identifier (DOI) such that it can be cited independently in the future. For instructions see: http://journals.plos.org/plosone/s/submission-guidelines#loc-laboratory-protocols

We look forward to receiving your revised manuscript.

Kind regards,

J Joe Hull, Ph.D.

Academic Editor

PLOS ONE

Journal Requirements:

2. We note that you are reporting an analysis of a microarray, next-generation sequencing, or deep sequencing data set. PLOS requires that authors comply with field-specific standards for preparation, recording, and deposition of data in repositories appropriate to their field. Please upload these data to a stable, public repository (such as ArrayExpress, Gene Expression Omnibus (GEO), DNA Data Bank of Japan (DDBJ), NCBI GenBank, NCBI Sequence Read Archive, or EMBL Nucleotide Sequence Database (ENA)). In your revised cover letter, please provide the relevant accession numbers that may be used to access these data. For a full list of recommended repositories, see http://journals.plos.org/plosone/s/data-availability#loc-omics or http://journals.plos.org/plosone/s/data-availability#loc-sequencing.

Reviewers' comments:

Reviewer's Responses to Questions

**Comments to the Author**

1. Is the manuscript technically sound, and do the data support the conclusions?

Reviewer #1: No

Reviewer #2: Yes

2. Has the statistical analysis been performed appropriately and rigorously? 

Reviewer #1: I Don't Know

Reviewer #2: Yes

3. Have the authors made all data underlying the findings in their manuscript fully available?

Reviewer #1: No

Reviewer #2: Yes

4. Is the manuscript presented in an intelligible fashion and written in standard English?

Reviewer #1: Yes

Reviewer #2: Yes

5. Review Comments to the Author

Reviewer #1: In this paper, the authors conducted RNAseq for C1 (H) and rep-1 silkworm strains 36hours after oviposition, and conducted qRT-PCR for with the same strains at the same stage.

I have two major concerns for this study.

One is that the authors are using the name rep-1 for the newly found egg color mutant without offering basis that it is an allele of the re mutant. They refer to a paper that this mutant is controlled by a recessive gene on the autosome, but the paper could not be easily accessed at least by google scholar search. Nor the information on the chromosome number the gene is located is provided.

To use the name rep-1, confirmation by complementation test with re is necessary to confirm that it is an allele of re (cross re with rep-1 and confirm that WT egg pigmentation does not occurr in the next generation. If the F1 eggs have WT color, rep-1 is not an allele of re. In this case, the authors should name this mutant with another name which is not used for already existing mutants.).

Another concern is that this paper needs more experiments and data before publishing in PLOS One.

I suggest the authors at add the following data.

Egg photos of C1 (H) and rep-1 strains at 36h and at least 1 time point each before and after 36h post oviposition.

photos of adult eyes of C1 (H) and rep-1 strains

complementation test with re mutant

If it is not an allele of re, linkage analysis for rep-1(RAD-seq using different WT strain may be effective)

conduct qRT-PCR for fig 6 or RNAseq for other stages (at the least, one point before 36h)

I have some other comments to improve the manuscript.

The first sentence of the abstract should be deleted, because it has little relevance with the main subject of this paper.

Figures 1, 2, 3 should be moved to Supplementary data.

line 66-68

"Eventually, two molecules of 3-hydroxy-kynurenine are synthesized by phenoxazinone synthetase (PHS) to produce lutein."

This is incorrect. The authors seem to mistaken xanthommatin for lutein. Lutein is carotenoid, not ommochrome.

line 64: "pyrrolasc" may be typo (unfamiliar term to me)

figure 6 (seems to be the most important figure at present): Presenting only NCBI gene ID is not common. The name of the gene or should be stated. If it is not a previously studied gene, at least kind of the gene, such as non-coding RNA, and the chromosome number the gene is located should be stated.

figure 7: please state clearly in the text whether the expression data for 12 and 24 hours post oviposition were conducted by the authors for this manuscript (If so, the results should in the results section and stated clearly in the materials and methods section) or refers to data from a previous study. Also, the label for vertical axis should be clearly stated (is this expression data of rep-1 or WT?)

Reviewer #2: I have 3 concerns in the Manuscript

1. In Materials and methods, line no 127 onwards, the numbering can be brought down, instead of running text as it confuses in continuous reading

2. Flow chart can be made for the various processes involved in Quality check of the data and its analysis

3. In references, the complete author names can be mentioned instead of et al like in 2nd, 8th, 11,12, 14, 15,20, 21, 32, 35 and 36.

6. PLOS authors have the option to publish the peer review history of their article (what does this mean?). If published, this will include your full peer review and any attached files.

Reviewer #1: No

Reviewer #2: Yes: A.H Manjunatha Reddy

---

## [Author Response · Author response to Decision Letter 0]

22 Apr 2020

Dear Editor and Reviewers:

 For our detailed response to this manuscript, please refer to the Response to Reviewer document for details. (As some Figures involved in the response which cannot be uploaded.) Thank you!

---

## [Decision Letter · Decision Letter 1]

21 May 2020

PONE-D-20-03586R1

Transcriptome analysis of the eggs of the silkworm pale red egg (rep-1) mutant at 36 hours after oviposition

PLOS ONE

Dear Dr. Zhao,

Thank you for submitting your revised manuscript to PLOS ONE. Although the revisions addressed many of mine and the Reviewer's comments, there remain a number of relatively minor points that could be more fully considered.

While I appreciate the author's direct replies, it would perhaps be more useful to the manuscript audience if the replies were incorporated directly into the manuscript. In addition, some confusion remains regarding the allele and its functionality. From my reading, I agree with the Reviewer that the color change would most likely be associated with a loss of pigmentation upstream. It would be most helpful if this was further clarified.

While the back crosses suggested by the Reviewer would undoubtedly provide further insights (and are greatly encouraged if possible), I think that by addressing the concerns discussed above regarding rep-1 should be sufficient for this study. Lastly, I agree with the Reviewer that the manuscript is better served by having photos of the egg phenotypes within the paper rather than as a supplementary figure.

We look forward to receiving your revised manuscript.

Kind regards,

J Joe Hull, Ph.D.

Academic Editor

PLOS ONE

Reviewers' comments:

Reviewer's Responses to Questions

**Comments to the Author**

1. If the authors have adequately addressed your comments raised in a previous round of review and you feel that this manuscript is now acceptable for publication, you may indicate that here to bypass the “Comments to the Author” section, enter your conflict of interest statement in the “Confidential to Editor” section, and submit your "Accept" recommendation.

Reviewer #1: (No Response)

2. Is the manuscript technically sound, and do the data support the conclusions?

Reviewer #1: Yes

3. Has the statistical analysis been performed appropriately and rigorously? 

Reviewer #1: I Don't Know

4. Have the authors made all data underlying the findings in their manuscript fully available?

Reviewer #1: Yes

5. Is the manuscript presented in an intelligible fashion and written in standard English?

Reviewer #1: Yes

6. Review Comments to the Author

Reviewer #1: 1)

From the authors information on the result of complementation test, I assume there is a possibility of rep-1 is an individual allele of re mutant with independent origin (though the coloration of F2 and F3 eggs may not be explained only by Bmre gene). However, this information is not provided in the revised text but only in the answers to reviewer and editor.

These are very important information and the original paper is not easily accessible to non-chinese researchers.

The genetic relationship between rep-1 and re (i.e. the results of complementation test), and also the mutation in Bmre gene in rep-1 mutant should be described and explained in the last paragraph of introduction section so that the general readers are given enough background information to evaluate the results of this study.

2)

Bm-re is not a transcription factor or a signal transducer, but a transporter. Since the authors stand on the basis that rep-1 is an allele of re and there is enough possility, they should discuss the link between the mutation in Bm-re gene in rep-1 mutant and the differential expression of genes between rep-1 and C1H, if they aim to say the differential expression of these genes are caused by Bm-re mutation.

However, there may be a possibility that differential gene expression detected in this study (i.e. Fig. 3) is caused by a locus other than Bm-re. Expression analysis between individuals obtained by several rounds of backcross may answer this.

3)

In the last sentence of introduction, the authors state that “The study aims to build a foundation for understanding the formation mechanism of rep-1 mutants and provides a theoretical basis for the study of the molecular mechanism of egg color formation.”

If we stand on the basis that rep-1 is an allele of re, the most simple explanation for color change in rep-1 is the lack of a pigment precursor due to the disruption of Bmre gene.

In the current version of the manuscript, the results of this study do not seem to add information on the cause of color change in the rep-1 mutant nor egg color formation.

4)

In the revised Figure 3, it is preferred that the name of the gene in addition to accession number is provided in the figure panel or in the figure legend.

5)

Photos of eggs are better included in the main figure than supplementary figure.

7. PLOS authors have the option to publish the peer review history of their article (what does this mean?). If published, this will include your full peer review and any attached files.

Reviewer #1: No

---

## [Author Response · Author response to Decision Letter 1]

3 Jun 2020

For more information, please check the "Response to Reviewers".

---

## [Editor Report · Decision Letter 2]

10 Jun 2020

PONE-D-20-03586R2

Transcriptome analysis of the eggs of the silkworm pale red egg (rep-1) mutant at 36 hours after oviposition

PLOS ONE

Dear Dr. Zhao,

Thank you for considering the Reviewer comments during the revision process. Although the comments/suggestions have been sufficiently addressed, because PLOS ONE does not utilize a copy editor, I would ask that you consider the edits (see attached) I have made to the manuscript that address a number of copyediting issues and/or unclear word usage. As such, we feel that although the manuscript has merit it does not yet fully meet PLOS ONE’s publication criteria as it currently stands. Therefore, we invite you to submit a revised version of the manuscript that addresses my suggested edits and questions.

We look forward to receiving your revised manuscript.

Kind regards,

J Joe Hull, Ph.D.

Academic Editor

PLOS ONE

---

## [Author Response · Author response to Decision Letter 2]

22 Jul 2020

Dear Editor and Reviewers:

Thank you for your valuable suggestions, we have answered the questions responses to each ponit raised by the academic editor and reviewers and made changes in the uploaded file ('Manuscript' and 'Revised Manuscript with Track Changes'). Please refer to the uploaded file for more details.

---

## [Editor Report · Decision Letter 3]

23 Jul 2020

Transcriptome analysis of the eggs of the silkworm pale red egg (rep-1) mutant at 36 hours after oviposition

PONE-D-20-03586R3

Dear Dr. Zhao,

We’re pleased to inform you that your manuscript has been judged scientifically suitable for publication and will be formally accepted for publication once it meets all outstanding technical requirements.

Kind regards,

J Joe Hull, Ph.D.

Academic Editor

PLOS ONE
---

## [Editor Report · Acceptance letter]

28 Jul 2020

PONE-D-20-03586R3 

Transcriptome analysis of the eggs of the silkworm pale red egg (rep-1) mutant at 36 hours after oviposition 

Dear Dr. Zhao:

I'm pleased to inform you that your manuscript has been deemed suitable for publication in PLOS ONE. Congratulations! Your manuscript is now with our production department. 

Kind regards, 

on behalf of

Dr. J Joe Hull 

Academic Editor

PLOS ONE